# Brief communication: Modulation instability of internal waves in a smoothly stratified shallow fluid with a constant buoyancy frequency

Kwok Wing Chow[1], Hiu Ning Chan[2], Roger H. J. Grimshaw[3]

[1]Department of Mechanical Engineering, University of Hong Kong, Pokfulam, Hong Kong
[2]Department of Mathematics, Chinese University of Hong Kong, Shatin, New Territories, Hong Kong
[3]Department of Mathematics, University College London, Gower Street, London, WC1E 6BT, United Kingdom

*Correspondence to*: K. W. Chow (kwchow@hku.hk)

**Abstract.** Unexpectedly large displacements in the interior of the oceans are studied through the dynamics of packets of internal waves, where the evolution is governed by the nonlinear Schrödinger equation. The case of constant buoyancy
frequency permits analytical treatment. While modulation instability for surface wave packets only arises for sufficiently deep water, 'rogue' internal waves may occur for the shallow water and intermediate depth regimes. The dependence on the stratification parameter and choice of internal modes can be demonstrated explicitly. The spontaneous generation of rogue waves is tested by numerical simulations.

## 1 Introduction

Rogue waves are unexpectedly large displacements from equilibrium positions or otherwise tranquil configurations. Oceanic rogue waves on the sea surface obviously pose immense risk to marine vessels and offshore structures (Dysthe, et al., 2008). After these waves were observed in optical waveguides, studies of such extreme and rare events have been actively pursued in many fields of science and engineering (Onorato et al., 2013). Within the realm of oceanic hydrodynamics, observation of rogue waves in coastal regions has been recorded (Nikolkina and Didenkulova, 2011; O'Brien et al., 2018). Nearly all
experimental and theoretical studies in the literature of rogue waves in fluids focus on surface waves. Our aim here is to investigate a similar scenario for internal waves. Internal waves play critical roles in the transport of heat, momentum and energy in the oceans, and breaking of such waves may have impact on circulation (Pedlosky, 1987). There is a quite substantial literature on observations and theories of large amplitude internal waves in shallow water (Stanton and Ostrovsky, 1998). Many studies concentrate on solitary waves in long wave situations employing the Korteweg-de Vries equation (Holloway et
al., 1997), but not on the highly transient modes with a potential of abrupt growth. For relevance in other fields of physics and engineering, the actual derivation of the governing equations may dictate the regime of input parameter values necessary for rogue waves to occur.

Theoretically the propagation of weakly nonlinear, weakly dispersive narrow-band wave packets is governed by the
nonlinear Schrödinger equation, where the dynamics is dictated by the competing effects of second order dispersion and cubic nonlinearity (Zakharov, 1968; Ablowitz and Segur, 1979). Modulation instability of plane waves and rogue waves can then occur only if dispersion and cubic nonlinearity are of the same sign. For surface wave packets on a fluid of finite depth, rogue

modes can emerge for $kh > 1.363$ where $k$ is the wavenumber of the carrier wave packet and $h$ is the water depth. Hence conventional understanding is that such rogue waves can only occur if the water depth is sufficiently large.

Other fluid physics phenomena have also been considered, such as the effects of rotation (Whitfield and Johnson, 2015) or the presence of shear current or an opposing current (Onorato et al., 2011; Toffoli et al., 2013a; Liao et al., 2017) or oblique perturbations (Toffoli et al., 2013b). While such considerations may change the numerical value of the threshold (1.363) and extend the instability region, the requirement of water of sufficiently large depth is probably unaffected. For wave packets of large wavelengths, dynamical models associated with the shallow water regime have been employed (Didenkulova and Pelinovsky, 2011, 2016), such as the well-known Korteweg-de Vries and Kadomtsev-Petviashvili types of equations (Grimshaw et al., 2010, 2015; Pelinovsky et al., 2000; Talipova et al., 2011), which may also lead to modulation instability under several special circumstances.

The goal here is to establish another class of rogue wave occurrence through the effects of density stratification, namely, internal waves in the interior of the oceans. Internal waves in general display more complex dynamical features than their surface counterparts. As an illustrative example, a given density profile may allow many internal modes characterized by the number of nodes in the vertical structures. This family of allowed states will be generically represented in this paper by an integer $n$ termed mode number. There is an extensive literature on large amplitude internal solitary waves which are spatially localized pulses propagating essentially without change of form (Grimshaw et al., 2004; Osborne, 2010). Our focus here is on internal rogue wave which is modelled as a wave pulse localized in **both** space and time. The asymptotic multiple scale expansions for internal wave packets under the Boussinesq approximation also yield the nonlinear Schrödinger equation (Grimshaw, 1977, 1981; Liu and Benney, 1981). When the buoyancy frequency is constant, modulation instability in one horizontal space dimension will only occur for $kh < k_c h = 0.766n\pi$ where the fluid is confined between rigid walls distance $h$ apart, $n$ is the vertical mode number of the internal wave, and the critical wave number $k_c$ given by (Liu et al., 2018):

$$k_c = \frac{n\pi}{h}\left(4^{1/3} - 1\right)^{1/2}. \tag{1}$$

For a wave packet associated with the first internal mode $(n = 1)$, modulation instability or rogue wave can occur for carrier wave number $k$ and shallow fluid of depth $h$ in the range of $kh < 0.766\pi$ or 2.406.

For a basin depth ($h$) of say 500m, the critical wavelength ($\lambda_c$) is

$$\lambda_c = \frac{2\pi}{k_c} = \frac{2h}{n(4^{1/3} - 1)^{1/2}}$$

and ranges of 'shallow' and 'intermediate' depths are covered (Table 1):

| $n$ (mode number of internal wave, with each $n$ representing a different vertical structure) | Rogue waves and instability can occur for wavelengths longer than $\lambda_c$ given by (in meters) |
|---|---|
| 1 | 1305 |
| 2 | 652 |
| 3 | 435 |
| 4 | 326 |
| 5 | 261 |

Table 1: Critical wavelength $\lambda_c$ as a function of various internal mode number $n$ (with $h = 500$ m).

    The important point is not just a difference in the numerical value of the cutoff, but rogue waves now occur for water depth **less** than a certain threshold. Our contribution is to extend this result. The nonlinear focusing mechanism of internal rogue waves is: (i) determined by estimation of the growth rate of modulation instability, and (ii) elucidated by a numerical simulation of emergence of rogue modes with the optimal modulation instability growth rate as the initial condition.

## 2 Formulation

### 2.1 Nonlinear Schrödinger theory for stratified shear flows

The dynamics of small amplitude (linear) waves in a stratified shear flow with the Boussinesq approximation is governed by the Taylor-Goldstein equation ($\phi(y)$ = vertical structure, $k$ = wavenumber, $c$ = phase speed, $U(y)$ = shear current):

$$\phi_{yy} - \left(k^2 + \frac{U_{yy}}{U - c}\right)\phi + \frac{N^2\phi}{(U - c)^2} = 0 \ , \tag{2}$$

where $N$ is the Brunt-Väisälä frequency or more simply 'buoyancy frequency' ($\bar{\rho}$ is the background density profile):

$$N^2 = -\frac{g}{\bar{\rho}}\frac{d\bar{\rho}}{dy} \ . \tag{3}$$

The evolution of weakly nonlinear, weakly dispersive wave packets is described by the nonlinear Schrödinger equation for the complex-valued wave envelope $S$, obtained through a multi-scale asymptotic expansion, which involves calculating the induced mean flow and second harmonic ($\beta$, $\gamma$ being parameters determined from the density and current profiles):

$$iS_\tau - \beta S_{\xi\xi} - \gamma|S|^2 S = 0 \ , \quad \tau = \varepsilon^2 t \ , \quad \xi = \varepsilon(x - c_g t) \tag{4}$$

where $\tau$ is the slow time scale, $\xi$ is the group velocity ($c_g$) coordinate and $\varepsilon$ is a small amplitude parameter.

### 2.2 Constant buoyancy frequency

    For the simple case of **constant** buoyancy frequency $N_0$, the formulations simplify considerably in the absence of shear flow ($U(y) = 0$). The linear theory Eq. (2) yields simple solutions for the mode number $n$:

$$N = N_0 \ , \quad \phi = \sin\left(\frac{n\pi y}{h}\right), \tag{5}$$

with the dispersion relation, phase velocity ($c$) and group velocity ($c_g$) given by

$$\omega^2 = \frac{k^2 N_0^2}{\frac{n^2\pi^2}{h^2} + k^2} \, , \qquad c = \frac{\omega}{k}, \qquad c_g = \frac{d\omega}{dk}, \qquad c_g = \frac{c}{1 + \frac{k^2 h^2}{n^2\pi^2}} \, . \tag{6}$$

The subsequent nonlinear analysis yields the coefficients of the nonlinear Schrödinger equation in explicit forms:

$$\beta = \frac{3n^2\pi^2 c^2}{2h^2 k N_0^2}\left(c - c_g\right), \qquad \gamma = -\frac{6N_0^2 k c_g^3\left(c - c_g\right)}{c^4\left(c^3 - 4c_g^3\right)} \, . \tag{7}$$

A plane wave solution for Eq. (4) (or physically a continuous wave background of amplitude $A_0$) is

$$S = A_0 \exp[-i\gamma A_0^2 \tau] \, . \tag{8}$$

Small disturbances with modal dependence $\exp[i(r\xi - \Omega\tau)]$ will exhibit modulation instability if

(a) $\Omega^2 = \beta r^2(\beta r^2 - 2\gamma A_0^2)$ is negative, i.e. for $\beta\gamma > 0$; calculations using Eqs. (6, 7) lead to $kh < k_c h = 0.766n\pi$ (Eq. (1));

(b) the maximum growth rate is (imaginary part of $\Omega$) = $\Omega_i = |\gamma|A_0^2$ for a special wavenumber given by $\beta^{1/2}r = \gamma^{1/2}A_0$;

(c) the growth rate for long wavelength disturbance is $|\Omega_i/r| = (2\beta\gamma)^{1/2}A_0$ for $r \to 0$.

      In terms of significance in oceanography, the constraint $kh < k_c h = 0.766n\pi$ does *not* depend on the constant buoyancy frequency $N_0$. However, it *does* depend on the mode number ($n$) of the internal wave, with the higher order modes permitting a large range of carrier envelope wavenumber and fluid depth for rogue waves to occur. An analysis in the long wave regime of this Taylor-Goldstein formulation would in principle recover the previous results related to the Korteweg-de Vries and

Gardner equations, and details will be reported in the future.

## 3. Computational Simulations

An intensively debated issue in the studies of rogue waves through a deterministic approach is the proper initial condition which may generate or favour the occurrence of such large amplitude disturbances. Modulation instability refers to the growth

of small disturbance in a system due to the interplay between dispersive and nonlinear effects (Craik, 1984), and here we examine this by solving the nonlinear Schrödinger equation (Eq. (4)) numerically. One suggestion is the role played by long wavelength modes associated with modulation instability, or 'baseband instability' (Baronio et al., 2015). To examine this effect and to clarify the role of stratification as well as the choice of internal wave modes, numerical simulations are performed where baseband modes with the maximum modulation instability growth rate on a plane wave background and say 5%

amplitude are selected as the initial condition (Chan and Chow, 2017; Chan et al., 2018):

$S(\xi,0) = [1 + 0.05\exp(ir\xi)]A_0$,

($A_0$ = the amplitude parameter defined by Eq. (8) and $r$ = wavenumber of the baseband mode).

This choice of a preferred modulation instability mode as the initial condition is different from other approaches in the literature, such as one using random noise. A pseudospectral method with a fourth-order Runge-Kutta scheme for marching forward in

time is applied to solve the nonlinear Schrödinger equation (Eq. (4)) numerically. When the wavenumber $r$ of the disturbance is small, corresponding to a baseband mode, rogue wave can be generated from the plane wave background (Figure 1).

Physically this spontaneous growth of disturbance due to modulation instability is closely associated with the 'focusing' of energy and thus the formation of rogue waves.

The growth rate of the baseband mode is a crucial factor of rogue wave generation. A stronger baseband growth rate will trigger a rogue wave within a shorter period of time. From Eqs. (6) and (7), the baseband growth rate $(2\beta\gamma)^{1/2}A_0$ increases as the depth $h$ or wavenumber $k$ increases (Figure 2), but this growth rate decreases as the mode number $n$ increases. However, this baseband rate is independent of the buoyancy frequency $N_0$. Numerical simulations were performed with parameter values appropriate for applications to fluid mechanics. The concrete numerical values of the growth rates in a laboratory frame of reference (time $t$) can be estimated from definitions used in Eq. (4), i.e. $\tau = \varepsilon^2 t$ and the small amplitude parameter $\varepsilon$ actually employed.

Figure 1 shows that rogue waves can emerge sooner when the fluid is deeper. Remarkably, this implies that baseband instability is stronger when the system is closer to the singular limit where the cubic nonlinearity changes sign. On the other hand, the degree of the background density stratification posts only a minor effect to the baseband mode. Apart from choosing a preferred baseband mode, another perspective taken in the literature is to select a random field as the initial condition. For the present nonlinear Schrödinger equation, 'rogue wave like' entities would then emerge too (Akhmediev et al., 2009).

## 4. Discussions and Conclusions

An analytically tractable model for packets of internal waves is studied through four input parameters, $h$ (fluid depth), $k$ (wavenumber of the carrier envelope packet), $N_0$ (buoyancy frequency), and $n$ (mode number of the internal wave), with only $h$ and $k$ relevant for surface waves. For internal waves, modulation instabilities and rogue waves may now arise for the shallow water and intermediate depth regimes if $N_0$ is constant. With knowledge of baseband instability and supplemented by computer simulations, the nonlinear focusing mechanism of rogue waves is assessed. Remarkably the constant buoyancy frequency may not play a critical role in the existence condition in terms of focusing, but the mode number of the internal wave does. For breathers or other pulsating modes, this buoyancy frequency parameter will enter the focusing mechanism consideration and further analytical and computational studies will be valuable (Sergeeva et al., 2014). In the next phase of this research effort, contrasts and similarities with surface waves should also be pursued, where a directional field or opposing currents can provide rogue waves generation mechanisms beyond the well established criterion of $kh > 1.363$. Such effects of shear currents and comparisons with experimental / field data will be taken up in future studies (Onorato et al. 2011; Toffoli et al. 2013a; Toffoli et al. 2013b). Density profiles with variable buoyancy frequency will also be examined in the future. Besides their relevance in transport phenomena, internal waves have significant connection with underwater acoustics (Apel et al, 2007; Zhou et al., 1991) and abnormally large internal rogue waves may have physical implications in those aspects.

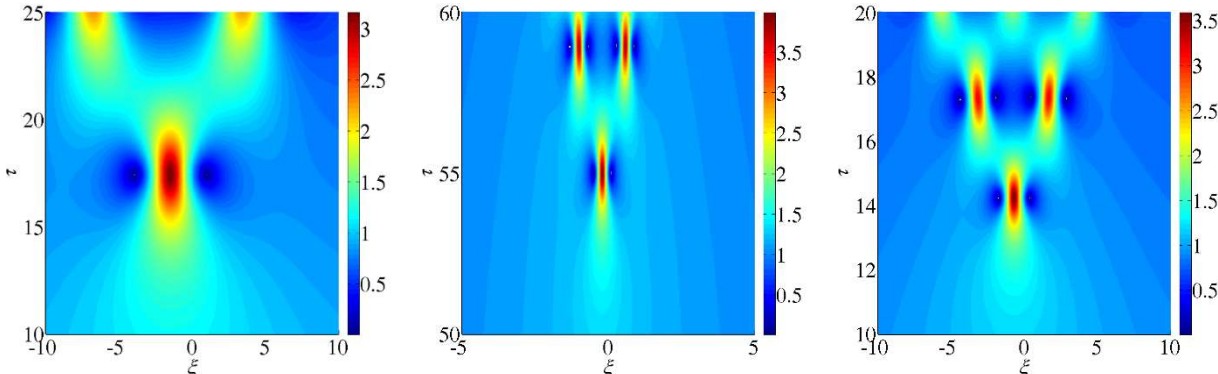

**Figure 1: The emergence of rogue wave modes from a background continuous wave perturbed by a long wavelength unstable mode. Larger baseband gain implies a smaller time is required for the rogue wave modes to emerge. Left: For $N_0 = 2$, $h = 4$, $k = 0.5$, $n = 1$, $r = 0.2$, and baseband instability growth rate = 0.868, rogue wave emerges at $\tau \approx 17$; Middle: For $N_0 = 2$, $h = 1$, $k = 0.5$, $n = 1$, $r = 0.2$, and baseband instability growth rate = 0.193, longer time is required for the emergence of rogue wave in a shallower fluid ($\tau \approx 55$); Right: For $N_0 = 1$, $h = 4$, $k = 0.5$, $n = 1$, $r = 0.2$, and baseband instability growth rate = 0.868, rogue wave emerges at about the same time ($\tau \approx 14$) as compared to the case with a higher buoyancy frequency $N_0 = 2$. In all cases, the amplitude of the background continous wave ($A_0$) is taken as 1.**

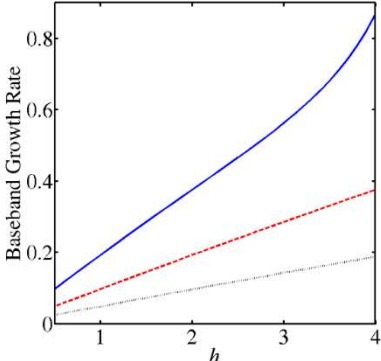

**Figure 2: The baseband growth rate increases as the fluid depth $h$ increases: $N_0 = 2$, $k = 0.5$, $n = 1$ (blue solid line); $N_0 = 2$, $k = 0.5$, $n = 2$ (red dashed line) ; $N_0 = 2$, $k = 0.25$, $n = 1$ (black dotted line).**

**Acknowledgements**

Partial financial support has been provided by the Research Grants Council (contracts HKU17200815 and HKU17200718).

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
