# Peer review of "Brief communication: Modulation instability of internal waves in a smoothly stratified shallow fluid with a constant buoyancy frequency"

_Natural Hazards and Earth System Sciences, 2018_

## Referee Comment (RC1) · E. Pelinovsky (Referee) · 16 Aug 2018

At the beginning, I would like to say that I liked the article due to its presentation clarity and the clarity of the results. The authors focused on internal rogue waves appearing due to modulation instability only and demonstrated modulation instability conditions for internal waves in the constant buoyancy ocean. Nevertheless, I have a few minor comments.

1. Taking into account the specificity of the journal (Natural Hazards. . ..), it would be reasonable to give an example of rogue wave characteristics in numbers using the formula (1), for instance, characteristic lengths of carrier and envelope waves in 100mdepth basin.

2. The authors state in Conclusion that "internal waves, modulation instabilities and rogue waves now occur for the shallow water regime". But this conclusion has been made earlier in the paper (Grimshaw, R., Pelinovsky, E., Talipova, T., and Sergeeva, A.: Rogue internal waves in the ocean: Long wave model, Eur. Phys. J. Special Topics, 185, 195–208, 2010) and in previous papers with Roger Grimshaw, where nonlinear Schrodinger equation has been derived from extended Korteweg-de Vries (Gardner) equation with positive cubic nonlinear term (see also Talipova T.G., Pelinovsky E.N., Kharif Ch. Modulation instability of long internal waves with moderate amplitudes in a stratified horizontally inhomogeneous ocean. JETP Letters, 2011, vol. 94, No. 3, 182-186). In the reviewed paper the nonlinear Schrodinger equation is derived from initial equations with full dispersion using Grimshaw's and his co-authors' previous papers. It means that author's criterion should include the positivity of cubic nonlinear term in Gardner as a particular case. Is it correct?

3. I think that the new important result is that the modulation instability cannot occur only in shallow water (according to the authors), but also in the intermediate depth basin.

4. Speaking of physical mechanisms of surface rogue wave formation in shallow water, perhaps, the following papers (Pelinovsky E., Talipova T., Kharif C. Nonlinear dispersive mechanism of the freak wave formation in shallow water. Physica D, 2000, vol. 147, No. 1-2, 83-94: Didenkulova I., and Pelinovsky E. Rogue waves in nonlinear hyperbolic systems (shallow-water framework). Nonlinearity, 2011, vol. 24, R1-R18) should be cited.

---

## Author Comment (AC1) · 16 Aug 2018

NHESS_2018_238 'The occurrence of rogue waves in the interior of the oceans: A modelling and computational study'
by K. W. Chow, H. N. Chan, and R. H. J. Grimshaw
Date: August 16, 2018

Reply to Referee 1:

We thank Professor Pelinovsky for his insightful and supportive comments. We respond in detail as follows.

(1) '*…it would be reasonable to give an example of rogue wave characteristics in numbers using formula (1), for instance, characteristic lengths of carrier and envelope waves in 100m-depth basin…*'

Response: Thank you for pointing out the need to verify the actual numerical orders of magnitude. For a basin depth ($h$) of say 500m, the critical wavelength ($\lambda_c$), wavenumber ($k_c$) and internal wave mode ($n$), the formulation in the text gives

$$\lambda_c = \frac{2\pi}{k_c} = \frac{2h}{n(4^{1/3} - 1)^{1/2}}$$

and with $h$ = 500m, we can construct the following table:

| $n$ (internal mode number) | Rogue waves and instability can occur for wavelengths longer than $\lambda_c$ given by (in meters) |
|:---:|:---:|
| 1 | 1305 |
| 2 | 652 |
| 3 | 435 |
| 4 | 326 |
| 5 | 261 |

Hence ranges of 'shallow' and 'intermediate' depths are covered. We will add this appropriate text. (Note: we change the suggested depth to 500m, to get a better approximation for the oceanic situation.)

(2) '*…rogue waves now occur for the shallow water regime…, but this conclusion has been made earlier in the paper…2010, and previous papers…2011…*'

Response: Thank you for reminding us of these works, which are certainly relevant and will be referenced in the revision stage. However, there is a subtle difference between the two approaches. In the previous works by one of the authors in 2010 and 2011, the starting point was a long wave model, the extended Korteweg-de Vries equation. There is thus an assumption of long waves in the basic carrier wave envelope. In contrast, the Taylor-Goldstein equation for linear modes is utilized in

the present approach, and hence the fast oscillations inside the carrier envelope need not be in the long wave regime.

'…*the author's criterion should include the positivity of cubic nonlinear term in (the) Gardner (equation) as a particular case. Is it correct?*'

Yes, if we take the small wavenumber regime for the Taylor-Goldstein equation, then we can recover the Korteweg-de Vries and Gardner equations. However, such an asymptotic calculation will take us way beyond the 4-page limit of a 'brief communication' paper. Nevertheless, we will mention this connection.

(3) '…*important result is that the modulation instability* *can occur not only* *in shallow water,…but also in the intermediate depth basin.*' (underline = our re-phrasing)

Response: Yes, that is exactly one of our messages in writing this paper and we will emphasize this point.

(4) '…*the following papers…should be cited.*'

Response: Thank you, we shall add citations of these papers.

---

## Referee Comment (RC2) · Anonymous Referee #2 · 31 Oct 2018

This manuscript discusses the formation of internal rogue waves. The topic is interesting for the audience of this journal and it is worth publication. Nevertheless, although this manuscript is prepared for a brief communication, it is too short to convene a significant and novel message that justifies a rapid communication. In my opinion, substantial work is needed before this manuscript can be considered for publication.

Specific comments:

1. Title is misleading. It mentions occurrence of rogue waves, which makes me think that formation of internal rogue waves is discussed within a proper statistical framework. However, this is not a case for the present manuscript.

2. There is an extensive literature discussing generation of internal rogue waves, but this is not discussed in details in the present manuscript. I am thinking, for example, to Grimshaw, R., Pelinovsky, E., Stepanyants, Y. and Talipova, T., 2006. Modelling internal solitary waves on the Australian North West Shelf. Marine and Freshwater Research, 57(3), pp.265-272; and Chapter 25 of Osborne, A.R., 2002. Nonlinear Ocean Wave and the Inverse Scattering Transform. In Scattering (pp. 637-666), and reference therein. To justify a rapid communication, more effort should be put to highlight the original contribution of the present manuscript.

3. The authors mention that "classical" modulation instability would cease at kh<1.36. However, there is evidence that instability can survive for shallower relative depth if the wave field is sufficiently directional (Toffoli, A., Fernandez, L., Monbaliu, J., Benoit, M., Gagnaire-Renou, E., Lefevre, J.M., Cavaleri, L., Proment, D., Pakozdi, C., Stansberg, C.T., Waseda, T., Onorato, M., 2013. Experimental evidence of the modulation of a plane wave to oblique perturbations and generation of rogue waves in finite water depth. Phys. Fluids, 25, 09170). Also, effects of current have been discussed in detail in Onorato M., Proment D., Toffoli A., 2011. Triggering rogue waves in opposing currents. Phys. Rev. Lett.,107, 184502, doi: 10.1103/PhysRevLett.107.184502; and Toffoli, A., Waseda, T., Houtani, H., Kinoshita, T., Collins, K., Proment, D., Onorato, M., 2013. Excitation of rogue waves in a variable medium: An experimental study on the interaction of water waves and currents. Phys. Rev. E, 87, 051201(R), before Liao et al 2017.

4. The theoretical framework, especially the NLS equation, seems to be already published. Nevertheless, the title mentions modelling study. What is the novel model the authors are proposing?

5. Section 3, Computational Simulations, is may major concern. It should be the core of the manuscript and yet it is reduced to 7 lines. This section does not convene a message at all and needs to be re-written and expanded.

6.What simulations did the author carried out? What are the initial conditions? Are regular or irregular waves considered? What are the values of key parameters? etc.. It also seems that no sensitivity analysis has been done and only one specific "lucky" case is discussed. What is the effect of wave steepness? What is the threshold of relative water depth below which internal rogue waves do not occur? what is the effect of density gradient? None of these points are discussed, leaving the reader completely unaware of the number computations. In addition, I am not sure to understand Figure 1. Or better, I can guess what it is and its meaning, but the authors did not put any effort to describe it.

7. Throughout the paper and in the title, it is mentioned that likelihood of occurrence of rogue waves is assessed. However, I do not see any discussion of a proper statistical framework that can justify new results on the probability of occurrence of internal rogue waves.

In conclusion, I think section 3 has to be significantly redeveloped and more details provided to support results. If this is done properly, this manuscript has the potential to become a significant contribution to ocean science.

---

## Author Comment (AC2) · 12 Nov 2018

NHESS_2018_238 'The occurrence of rogue waves in the interior of the oceans: A modelling and computational study'
by K. W. Chow, H. N. Chan, and R. H. J. Grimshaw
Date: November 12, 2018

Reply to Referee 2:

We thank the Referee for the constructive comments, and also for the assertion that the present work can be potentially an important contribution to ocean science. Before we submit a revised manuscript, we provide a few concise points in response:

(1) '… *occurrence of rogue waves, which makes me think that formation of internal rogue waves is discussed within a proper statistical framework*…'

Response: There are several review articles on rogue waves where various approaches of investigations are presented including deterministic and stochastic models. An example is 'Oceanic rogue waves' by K. Dysthe, H. E. Krogstad, P. Muller, *Annual Reviews of Fluid Mechanics* **40**, 287 (2008). From the various mechanisms discussed, the words 'nonlinear focusing' and 'modulational instability' are more appropriate for our paper. Hence we suggest a possible change to a new title 'Nonlinear focusing as a possible generation mechanism for rogue waves in the interior of the oceans: A modelling and computational study'.

(2) '…*There is an extensive literature discussing generation of internal rogue waves, but this is not discussed in details in the present manuscript. I am thinking, for example, to Grimshaw, R., Pelinovsky, E., Stepanyants, Y. and Talipova, T., 2006. Modelling internal solitary waves on the Australian North West Shelf. Marine and Freshwater Research, 57(3), pp.265-272; and Chapter 25 of Osborne, A.R., 2002. Nonlinear Ocean Wave and the Inverse Scattering Transform. In Scattering (pp. 637-666), and reference therein. To justify a rapid communication, more effort should be put to highlight the original contribution of the present manuscript*…'

Response: There is indeed an extensive literature on large amplitude oceanic internal waves. In particular, the two references quoted and many other related works are mainly on the topic of 'internal solitary waves'. These are spatially localized pulses propagating permanently, but they are not localized in time. We consider simple analytical description of a wave pulse localized in **both** space and time. In widely used phrase in this field, rogue waves are 'waves that appear from nowhere and disappear without a trace'. We shall emphasize on this point in the revised manuscript.

(3) '*…The authors mention that "classical" modulation instability would cease at kh<1.36. However, there is evidence that instability can survive for shallower relative depth if the wave field is sufficiently directional (Toffoli, A., Fernandez, L., Monbaliu, J., Benoit, M., Gagnaire-Renou, E., Lefevre, J.M., Cavaleri, L., Proment, D., Pakozdi, C., Stansberg, C.T., Waseda, T., Onorato, M., 2013. Experimental evidence of the modulation of a plane wave to oblique perturbations and generation of rogue waves in finite water depth. Phys. Fluids, 25, 09170). Also, effects of current have been discussed in detail in Onorato M., Proment D., Toffoli A., 2011. Triggering rogue waves in opposing currents. Phys. Rev. Lett.,107, 184502, doi: 10.1103/PhysRevLett.107.184502; and Toffoli, A., Waseda, T., Houtani, H., Kinoshita, T., Collins, K., Proment, D., Onorato, M., 2013. Excitation of rogue waves in a variable medium: An experimental study on the interaction of water waves and currents. Phys. Rev. E, 87, 051201(R), before Liao et al 2017…*'

Response: Thank you for these references, where the well-known constraint of $kh >$ 1.363 was extended to lower numerical values. However, it is not clear (at least to us) how far can these numerical values go. In contrast,
► we are studying internal waves as opposed to surface waves, and
► our proposed constraint is very well defined, i.e. $kh < 0.766n\pi$. The limit of $k$ or $h$ tending is zero is explicitly included.

(4) '*…The theoretical framework, especially the NLS equation, seems to be already published. Nevertheless, the title mentions modelling study. What is the novel model the authors are proposing?...*'

Response: The word 'modelling' is used here as opposed to numerical simulations or field data comparison. When the paper by Liu and Benney (*Studies in Applied Mathematics* 1981) was published, the focus then was internal solitary wave. Our proposed contributions are:

(a) This formulation applied to the setting of internal rogue waves will provide a nonlinear focusing mechanism in the long internal waves (shallow water) regime, as opposed to the usual deep water scenario for surface waves.

(b) Numerical simulations from random and specially prescribed initial conditions, a practice frequently implemented only in the past ten years, is pertinent for internal wave investigations.

(5) '…Section 3, *Computational Simulations, is may major concern. It should be the core of the manuscript and yet it is reduced to 7 lines. This section does not convene a message at all and needs to be re-written and expanded*…'

Response: Please see point (6) below for a full explanation.

(6) We provide a response to each query individually. As an overview, the primary intention of this 'brief communication' is to demonstrate that unexpectedly large displacements (rogue waves) may occur in internal waves too. Indeed they can occur in the shallow water regime, in sharp contrast to the surface wave scenarios. Numerical presentations were condensed due to the 4-page limit, but we can expand this part if necessary, subject to editorial advice.

'…*What simulations did the author carried out?*…'
Response: We conduct simulations with random as well as specially selected initial conditions to determine how rogue-wave-like structures can emerge.

'…*What are the initial conditions? Are regular or irregular waves considered?*…'
Response: Specially selected conditions mean choosing a modulation instability mode with the optimized growth rate. Random conditions are generated by the computer.

'…*What are the values of key parameters? etc*…'
Response: For surface rogue waves described by the nonlinear Schrödinger equation, the key parameters are $k$, the wave number of the carrier wave envelope and $h$, the water depth. For the present wave packet dynamics in a stratified flow model, two additional parameters are $N_0$, the constant buoyancy frequency of the background stratification and $n$, the mode number of the internal wave.

'…*It also seems that no sensitivity analysis has been done and only one specific "lucky"case is discussed*…'
Response: Standard quality control processes were routinely performed for similar simulations in our papers in the past. Our present results, analogous to those of other research groups (e.g. Baronio et al, *Physical Review A* 2015), are that rogue-wave-like structures will emerge, and this is not a 'lucky' result. The goal of this portion of the paper is to convince the audience that such dynamics also holds true for internal wave scenarios too. We would be pleased to revise our paper to provide more detailed discussions.

'…*What is the effect of wave steepness? What is the threshold of relative water depth below which internal rogue waves do not occur? what is the effect of density gradient?*…'
Response: The wave steepness must scale with the small parameter describing the long modulation scale as given in any standard derivation of the nonlinear

Schrödinger equation (e.g. the paper by Liu and Benney, *Studies in Applied Mathematics*, 1981). The threshold of relative water depth for ***internal rogue waves to occur*** is $kh < 0.766n\pi$, four lines below Equation (7) of the text (strong contrast with $kh > 1.363$ of surface waves – this constitutes the theme of the paper). This new constraint means that internal rogue waves can thus occur for small $h$ (or shallow water regime). The density gradient, or more precisely, the buoyancy frequency parameter $N_0$, will affect the horizontal length scale of the rogue wave and a precise description will constitute one of the long term objectives of this study.

'…*None of these points are discussed, leaving the reader completely unaware of the number computations. In addition, I am not sure to understand Figure 1. Or better, I can guess what it is and its meaning, but the authors did not put any effort to describe it…*'
Response: Again we wish to emphasize that we are constrained by the 4-page limit of a 'brief communication'. To address a relatively broad audience, we have described the dynamics of the nonlinear Schrödinger equation in the first half of the paper. If space permits, we can elucidate the numerical details in a revised paper if necessary.

(7) '…*Throughout the paper and in the title, it is mentioned that likelihood of occurrence of rogue waves is assessed. However, I do not see any discussion of a proper statistical framework that can justify new results on the probability of occurrence of internal rogue waves…*'

Response: As discussed earlier, it is beyond the scope of this paper to carry out a statistical assessment. To avoid confusion, we shall adopt the words 'nonlinear focusing mechanism' or similar terminology in the revised paper.

(8) Final paragraph:
'…*section 3 has to be significantly redeveloped and more details provided to support results…*'

Response: Again the motivation of writing this 'brief communication' is to show this rather unexpected parameter regime for nonlinear focusing for internal rogue waves. Due to the 4-page limit on a 'brief communication', we have of necessity condensed the numerical treatment. We can revise and expand the simulation portions, subject to editorial approval.

'…*If this is done properly, this manuscript has the potential to become a significant contribution to ocean science…*'

Response: Thank you for providing a very positive opinion on our work.

---

## Author Response (AR1)

**THE UNIVERSITY OF HONG KONG**

January 12, 2019

**The Editorial Board**
**Natural Hazards and Earth System Sciences**

Dear Editors of Natural Hazards and Earth System Sciences,

We are pleased to submit the revised version of the manuscript, **nhess-2018-238**, by Kwok Wing Chow, Hiu Ning Chan and Roger H. J. Grimshaw. We have changed the title of the paper as part of the response to the requests of one referee. All modifications implemented are highlighted in BLUE and underlined in the revised manuscript. A detailed point-by-point response to the concerns raised by the referees is documented in the following pages. We wish to thank the referees for their valuable opinions and also the Editorial Office for handling the manuscript.

Please feel free to contact us if you need further information. Thank you.

Yours sincerely,

KWC

Dr. K. W. Chow, kwchow@hku.hk
Professor, Department of Mechanical Engineering
University of Hong Kong

**nhess-2018-238**
Modified title: Modulation instability as a generation mechanism for internal oceanic rogue waves: A modelling and computational study
by Kwok Wing Chow, Hiu Ning Chan and Roger H. J. Grimshaw.

**Reply to Referee 1**:

  We wish to thank Referee 1 for his insightful and supportive comments. We respond in detail as follows.

(1) '…*it would be reasonable to give an example of rogue wave characteristics in numbers using formula (1), for instance, characteristic lengths of carrier and envelope waves in 100m-depth basin…*'

Response: Thank you for pointing out the need to verify the actual numerical orders of magnitude. For a basin depth ($h$) of say 500m, the critical wavelength ($\lambda_c$), wavenumber ($k_c$) and internal wave mode ($n$), the formulation in the text gives

$$\lambda_c = \frac{2\pi}{k_c} = \frac{2h}{n(4^{1/3}-1)^{1/2}}$$

and with $h$ = 500m, we can construct the following table:

| $n$ (internal mode number) | Rogue waves and instability can occur for wavelengths longer than $\lambda_c$ given by (in meters) |
| --- | --- |
| 1 | 1305 |
| 2 | 652 |
| 3 | 435 |
| 4 | 326 |
| 5 | 261 |

Hence ranges of 'shallow' and 'intermediate' depths are covered. This information has been added in page 2 (after Equation (1)) of the revised text. (Note: we change the suggested depth to 500m, to get a better approximation for the oceanic situation.)

(2) '…*rogue waves now occur for the shallow water regime…, but this conclusion has been made earlier in the paper…2010, and previous papers…2011…*'

Response: Thank you for reminding us of these relevant works on the long wave (shallow water) regime. However, there is a subtle difference between the two approaches. In the previous work by one of the authors (RHJG) in 2010, the starting

point was a long wave model, the extended Korteweg-de Vries equation. There is thus an assumption of long waves in the basic carrier wave envelope. In contrast, the Taylor-Goldstein equation for linear modes is utilized in the present approach, and hence the fast oscillations inside the carrier envelope need not be in the long wave regime. We have enhanced the connection to this body of works in the literature by incorporating these new references:

Didenkulova, I. and Pelinovsky, E.: Rogue wave in nonlinear hyperbolic systems (shallow-water framework), Nonlinearity, 24, R1-R18, 2011.

Pelinovsky, E., Talipova, T., and Kharif, Ch.: Nonlinear-dispersive mechanism of the freak wave formation in shallow water, Physica D, 147, 83-94, 2000.

Talipova, T. G., Pelinovsky, E. N., and Kharif, Ch.: Modulation instability of long internal waves with moderate amplitudes in a stratified horizontally inhomogeneous ocean, JETP Letters, 94, 182-186, 2011.

'…*the author's criterion should include the positivity of cubic nonlinear term in (the) Gardner (equation) as a particular case. Is it correct?*'

Yes, if we take the small wavenumber regime for the Taylor-Goldstein equation, then we can recover the Korteweg-de Vries and Gardner equations. However, such an asymptotic calculation will take us way beyond the 4-page limit of a 'brief communication' paper, and thus only a brief remark is made at the end of Section 2.

(3) '…*important result is that the modulation instability* *can occur not only* *in shallow water,…but also in the intermediate depth basin*.' (underline = our re-phrasing)

Response: Yes, that is exactly one of our messages in writing this paper and we will emphasize this point, both in the Abstract (line 4 of that paragraph) as well as Section 4 Discussions and Conclusions (line 4 of that section).

(4) '…*the following papers…should be cited*.'

Response: Thank you. The Physica D 2000 and Nonlinearity 2011 papers have been included in the References.

**nhess-2018-238**
Modified title: Modulation instability as a generation mechanism for internal oceanic rogue waves: A modelling and computational study
by Kwok Wing Chow, Hiu Ning Chan and Roger H. J. Grimshaw.

**Reply to Referee 2**:

We thank the Referee for the constructive comments, and also for the assertion that the present work can be potentially an important contribution to ocean science.

(1) '*Title is misleading… occurrence of rogue waves, which makes me think that formation of internal rogue waves is discussed within a proper statistical framework…*'

Response: There are several widely cited review articles on rogue waves where various approaches of investigations are presented, including both deterministic and stochastic models. An example is 'Oceanic rogue waves' by K. Dysthe, H. E. Krogstad, P. Müller, *Annual Reviews of Fluid Mechanics* **40**, 287 (2008). From the various mechanisms discussed, and to avoid the possible confusion associated with the word 'occurrence', the terminologies of 'nonlinear focusing' and 'modulation instability' are perhaps more appropriate for our paper. Hence we suggest a possible change to a new title 'Modulation instability as a generation mechanism for internal oceanic rogue waves: A modelling and computational study'.

(2) '*…There is an extensive literature discussing generation of internal rogue waves, but this is not discussed in details in the present manuscript. I am thinking, for example, to Grimshaw, R., Pelinovsky, E., Stepanyants, Y. and Talipova, T., 2006. Modelling internal solitary waves on the Australian North West Shelf. Marine and Freshwater Research, 57(3), pp.265-272; and Chapter 25 of Osborne, A.R., 2002. Nonlinear Ocean Wave and the Inverse Scattering Transform. In Scattering (pp. 637-666), and reference therein. To justify a rapid communication, more effort should be put to highlight the original contribution of the present manuscript…*'

Response: There is indeed an extensive literature on large amplitude oceanic internal waves. In particular, the two references quoted and many other related works are mainly on the topic of 'internal solitary waves'. These are spatially localized pulses propagating essentially without change of form, but they are **not** localized in time. In this paper we consider simple analytical description of a wave pulse localized in **both** space and time. In widely used phrase in this field, rogue waves are 'waves that appear from nowhere and disappear without a trace'. We emphasize on this

difference in page 2 of the revised manuscript (7 lines above Equation (1)), and nevertheless have included some relevant references.

(3) '…*The authors mention that "classical" modulation instability would cease at kh<1.36. However, there is evidence that instability can survive for shallower relative depth if the wave field is sufficiently directional (Toffoli, A., Fernandez, L., Monbaliu, J., Benoit, M., Gagnaire-Renou, E., Lefevre, J.M., Cavaleri, L., Proment, D., Pakozdi, C., Stansberg, C.T., Waseda, T., Onorato, M., 2013. Experimental evidence of the modulation of a plane wave to oblique perturbations and generation of rogue waves in finite water depth. Phys. Fluids, 25, 09170). Also, effects of current have been discussed in detail in Onorato M., Proment D., Toffoli A., 2011. Triggering rogue waves in opposing currents. Phys. Rev. Lett.,107, 184502, doi: 10.1103/PhysRevLett.107.184502; and Toffoli, A., Waseda, T., Houtani, H., Kinoshita, T., Collins, K., Proment, D., Onorato, M., 2013. Excitation of rogue waves in a variable medium: An experimental study on the interaction of water waves and currents. Phys. Rev. E, 87, 051201(R), before Liao et al 2017…*'*

Response: Thank you for these references, where the well-known constraint of $kh > 1.363$ was extended to lower numerical values. However, it is not clear (at least to us) how far can these numerical values go. In contrast,
► we are studying internal waves as opposed to surface waves, and
► our proposed constraint is very well defined, i.e. $kh < 0.766n\pi$. The limit of $k$ or $h$ tending to zero is explicitly included.

The effects of ocean / shear currents will be taken up in future studies. Experimental verification will be beyond the scope of the present study. Nevertheless we have made relevants remarks, mentioned all three papers in Section 4 and included them in the References section.

(4) '…*The theoretical framework, especially the NLS equation, seems to be already published. Nevertheless, the title mentions modelling study. What is the novel model the authors are proposing?…*'

Response: The word 'modelling' is used here as opposed to numerical simulations or field data comparison. When the paper by Liu and Benney (*Studies in Applied Mathematics* 1981) was published, the focus then was internal solitary wave. Our proposed contributions are:

(a) This formulation as applied to the setting of internal rogue waves will provide a nonlinear focusing mechanism in the long internal waves (shallow water) regime, as opposed to the usual deep water scenario for surface waves.

(b) Numerical simulations from random and specially prescribed initial conditions, a practice frequently implemented only in the past ten years, is pertinent for internal wave investigations.

(5) '…Section 3, *Computational Simulations, is may major concern. It should be the core of the manuscript and yet it is reduced to 7 lines. This section does not convene a message at all and needs to be re-written and expanded*…' (our guess: *my major concern*?)

Response: Please see point (6) below for a full explanation.

(6) We provide a response to each query individually. As an overview, the primary intention of this 'brief communication' is to demonstrate that unexpectedly large displacements (rogue waves) may occur in internal waves too. Indeed they can occur in the shallow water regime, in sharp contrast to the surface wave scenarios. Numerical presentations were condensed in the initial submission due to the 4-page limit. We have substantiated the contents in this revised version, and we can expand this part further if necessary, subject to editorial advice.

'…*What simulations did the author carried out?*...'
Response: We conduct simulations with specially selected initial conditions to determine how rogue-wave-like structures can emerge. More precisely, we choose a mode with an optimal modulation instability growth rate.

'…*What are the initial conditions? Are regular or irregular waves considered?*...'
Response: Specially selected conditions mean choosing a modulation instability mode with the optimized (or maximum) growth rate. Hence we can roughly classify them as 'regular waves'. Numerical simulations for the nonlinear Schrödinger equation with random initial conditions had been conducted earlier in the literature (Akhmediev et al., *Physical Review E* 2009, cited in the manuscript).

'…*What are the values of key parameters? etc*…'
Response: For surface rogue waves described by the nonlinear Schrödinger equation, the key parameters are $k$, the wave number of the carrier wave envelope and $h$, the water depth. For the present wave packet dynamics in a stratified flow model, two

additional parameters are $N_0$, the constant buoyancy frequency of the background stratification and $n$, the mode number of the internal wave.

'…*It also seems that no sensitivity analysis has been done and only one specific "lucky"case is discussed*…'
Response: Standard quality control processes were routinely performed for similar simulations in our papers in the past. Our present results, analogous to those of other research groups (e.g. Baronio et al, *Physical Review A* 2015), are that rogue-wave-like structures will emerge, and this is **not** a 'lucky' result. The goal of this portion of the paper is to convince the reader that such dynamics also holds true for internal wave scenarios too. We have expanded Section 3 by providing highlights of the computational schemes and can substantiate with further details, depending on editorial advice on the classification as a 'brief communication' versus 'full paper'.

'…*What is the effect of wave steepness? What is the threshold of relative water depth below which internal rogue waves do not occur? what is the effect of density gradient?*…'
Response: The wave steepness must scale with the small parameter describing the long modulation scale as given in any standard derivation of the nonlinear Schrödinger equation (e.g. the paper by Liu and Benney, *Studies in Applied Mathematics*, 1981, amongst many others). The threshold of relative water depth for **internal rogue waves to occur** is $kh < 0.766n\pi$, four lines below Equation (7) of the text (strong contrast with $kh > 1.363$ of surface waves – this constitutes the theme of the paper). This new constraint means that internal rogue waves can thus occur for small $h$ (or shallow water regime). The density gradient, or more precisely, the buoyancy frequency parameter $N_0$, will affect the horizontal length scale of the rogue wave and a precise description will constitute one of the long term objectives of this study.

'…*None of these points are discussed, leaving the reader completely unaware of the number computations. In addition, I am not sure to understand Figure 1. Or better, I can guess what it is and its meaning, but the authors did not put any effort to describe it*…'
Response: Again we wish to emphasize that we are constrained by the 4-page limit of a 'brief communication' in the initial submission. To address a relatively broad audience, we have described the dynamics of the nonlinear Schrödinger equation in the first half of the paper. We have included descriptions of numerical schemes in Section 3 now, and can elucidate the numerical details in a full paper if necessary. The caption of Figure 1 has been expanded to 6 lines, hopefully the science is more comprehensible now.

(7) '…*Throughout the paper and in the title, it is mentioned that likelihood of occurrence of rogue waves is assessed. However, I do not see any discussion of a proper statistical framework that can justify new results on the probability of occurrence of internal rogue waves…*'

Response: As discussed earlier, it is beyond the scope of this paper to carry out a statistical assessment. To avoid possible confusion with phrases like 'likelihood' or 'occurrence', we shall adopt the words 'modulation instability' in the modified title and also discussions in Section 4.

(8) Final paragraph:
'…*section 3 has to be significantly redeveloped and more details provided to support results…*'

Response: Again the motivation of writing this 'brief communication' is to show this rather unexpected parameter regime for the modulation instability of internal rogue waves. Due to the 4-page limit on a 'brief communication' in the initial submission, we have of necessity condensed the numerical treatment. We beef up the simulation portions already and can further expand on those treatments, subject to editorial approval.

'…*If this is done properly, this manuscript has the potential to become a significant contribution to ocean science…*'

Response: Thank you for providing a very positive opinion on our work.

[revised manuscript text omitted]

---

## Referee Report (RR1)

**Referee's Report**
**on the paper by Kwok Wing Chow, Hiu Ning Chan, and Roger H. J. Grimshaw,**
**Modulation instability as a generation mechanism for internal oceanic rogue waves: A modelling and computational study**

I received for review, apparently, the revised version of the paper, wheher the majority of questions have been addressed. The paper in its current form looks fine, interesting, and is acceptable for publication in the journal NHESS. However, there are few minor issues which should be further addressed to make the paper clearer for readers. There are:

1.  The paper title sound too general, whereas the authors in this brief communication report an interesting discovery that the modulation instability of internal waves in a smoothly stratified fluid with a constant buoyancy frequency can occur in the shallow-water limit, when $kh < 0.766n\pi$, in contrast with surface waves, where the modulation instability can occur when $kh > 1.363$. I would suggest to replace the title with something like this: "Modulation instability of internal waves in a smoothly stratified shallow water with a constant buoyancy frequency". The relevance of the paper to rogue waves can be announced in the abstract, not necessarily in the title.

2.  The authors present the nonlinear Shrodinger (NLS) equation for internal waves without derivation. This looks fine for the brief communication, however I would emphasise somewhere that it is assumed that the boundary conditions on the water surface is assumed in the fully nonlinear form. Namely this leads to the nonzero nonlinear coefficient in the NLS equation, because the hydrodynamic equations for internal waves with a constant buoyancy frequency are linear.

3.  In this paper the authors do not use the full version of Taylor–Goldstein equation with the mean shear flow, but only its reduced version when there is no flow. I am suggesting to replace Eq. (2) with the standard equation for internal waves.

4.  In Section 3 it would be useful to mention that the authors solve numerically the NLS equation (4).

5.  It is not clear, what are the dimensions of parameters in Figs. 1 and 2? It woul be good also to have an estimate for the dimensional maximumal growth rate of modulation instability for the first few modes and given buoyancy frequency for $h = 500$ m.

Yury Stepanyants.

---

## Author Response (AR2)

**THE UNIVERSITY OF HONG KONG**

February 28, 2019

**The Editorial Board**
**Natural Hazards and Earth System Sciences**

Dear Editors of Natural Hazards and Earth System Sciences,

Thank you for your email of February 21, 2019. We are pleased to submit the revised version of the manuscript, **nhess-2018-238**, by Kwok Wing Chow, Hiu Ning Chan and Roger H. J. Grimshaw. All modifications implemented in response to comments of the referees are highlighted in BLUE and underlined in the revised manuscript. The title has been modified to 'Modulation instability of internal waves in a smoothly stratified shallow fluid with a constant buoyancy frequency', in accordance to a suggestion from a referee. A detailed point-by-point response to the concerns raised by the referees is documented in the following pages. We wish to thank the referees for their valuable opinions and also the Editorial Office for handling the manuscript.

Please feel free to contact us if you need further information. Thank you.

Yours sincerely,

KWC

Dr. K. W. Chow, kwchow@hku.hk
Professor, Department of Mechanical Engineering
University of Hong Kong

**nhess-2018-238**
Modified title: Modulation instability of internal waves in a smoothly stratified shallow fluid with a constant buoyancy frequency
by Kwok Wing Chow, Hiu Ning Chan and Roger H. J. Grimshaw.

**Reply to Referee 3**:

We wish to thank Referee 3 for his insightful and supportive comments. We respond in detail as follows.

(1) '…*to replace the title with something like this: "Modulation instability of internal waves in a smoothly stratified shallow water with a constant buoyancy frequency"*…'

Response: This suggestion is adopted, except that the word 'water' is changed to 'fluid' to allow for a more generalized setting. Thank you.

(2) '*The authors present the nonlinear Shrodinger (NLS) equation for internal waves without derivation. This looks fine for the brief communication, however I would emphasise somewhere that it is assumed that the boundary conditions on the water surface is assumed in the fully nonlinear form. Namely this leads to the nonzero nonlinear coefficient in the NLS equation, because the hydrodynamic equations for internal waves with a constant buoyancy frequency are linear*.'

Response: The full derivation of the nonlinear Schrödinger equation governing the evolution of wave packets for a stratified shear flow was given in several earlier works, e.g., Grimshaw 1977, 1981, as well as Liu and Benney, 1981, as cited in the text. The fluid equations for internal waves with a constant buoyancy frequency are linear for plane harmonic waves only. For wave packets there is a wave-induced mean flow, which feeds back to generate a nonlinear term in the asymptotic development. A standard sequence of perturbation calculations and solvability condition will yield the nonlinear Schrödinger equation.

(3) '…*the full version of Taylor–Goldstein equation…suggesting to replace Eq. (2) with the standard equation for internal waves*…'

Response: The form given in the revised manuscript should be the full Taylor-Goldstein equation with shear flow and the buoyancy term.

(4) '…*useful to mention that the authors solve numerically the NLS equation (4)…*'

Response: The following statements were added at the beginning of Section 3:

Modulation instability refers to the growth of small disturbance in a system due to the interplay between dispersive and nonlinear effects (Craik, 1984), and here we examine this by solving the nonlinear Schrödinger equation (Eq. (4)) numerically.

A pseudospectral method with a fourth-order Runge-Kutta scheme for marching forward in time is applied to solve the nonlinear Schrödinger equation (Eq. (4)) numerically.

(5) '…*It is not clear, what are the dimensions of parameters in Figs. 1 and 2? It would be good also to have an estimate for the dimensional maximal growth rate of modulation instability for the first few modes and given buoyancy frequency for h = 500 m…*'

Response: The parameters are dimensionless in Figures 1 and 2 because we are conducting the analysis in a non-dimensional framework. Just like any standard derivation of the nonlinear Schrödinger equation in fluid mechanics, the propagation variable and transverse variable of Eq. (4) are slow time ($\varepsilon^2 t$) and group velocity coordinate. The actual growth pattern of modulation instability thus goes like

Exp[(growth rate) $\varepsilon^2 t$], where $t$ is the time measured in ordinary sense in a laboratory.

Typically a perturbation theory is applied for say $\varepsilon = 0.03$. With a growth rate of order one, the time for the amplitude to amplify by a factor of $e$ (or 2.718 numerically) would be roughly 1,000 seconds, i.e. 16.7 minutes. This is consistent with the time scale of oceanic internal waves. The actual magnitude of velocity can be estimated from $h = 500$ m and the given $\varepsilon$ of the perturbation theory. Note that baseband growth rate should be scaled by the wavenumber $r$ of the long wave disturbance. The following statement is added in Section 3: The concrete numerical values of the growth rates in a laboratory frame of reference (time $t$) can be estimated from definitions used in Eq. (4), i.e. $\tau = \varepsilon^2 t$ and the small amplitude parameter $\varepsilon$ actually employed.

**nhess-2018-238**
Modified title: Modulation instability of internal waves in a smoothly stratified shallow fluid with a constant buoyancy frequency
by Kwok Wing Chow, Hiu Ning Chan and Roger H. J. Grimshaw.

**Reply to Referee 2**:

We wish to thank Referee 2 for his opinions.

(1) '…*In answering my previous comment (3), the authors state that they are studying "internal rogue waves as opposed to surface waves". Yet, the introduction is strongly focussed on surface rogue waves. This is quite confusing and led to my suggestions for missing references*…'

Response: In contrast to surface rogue waves, there is very little work on this topic of possible rogue waves in the interior of the ocean. The goal of the present work is to propose one such plausible framework. As pointed out in the January 2019 (i.e. previous) revised manuscript, internal solitons may have a large literature but they are ***not*** rogue waves, as a soliton is not localized in time. Nevertheless, we have included some references for internal waves in general in the background discussion (Section 1).

'…*The authors should focus more on the main topic of their work and highlight the new and striking results they have achieved*…'

Response: Our theme of internal rogue waves occurring for the shallow water (long wavelength) regime has been emphasized at several places of the text.

'…*In addition, if one of the goals is to reach a broad audience,… beneficial to briefly explain potential relevance of this work in other fileds of physics or engineering*…'

Response: Brief remarks on the importance of internal waves are made and a few references in this field are added (Section 1):

Nearly all experimental and theoretical studies in the literature of rogue waves in fluids focus on surface waves. Our aim here is to investigate a similar scenario for internal waves. Internal waves play critical roles in the transport of heat, momentum and energy in the oceans, and breaking of such waves may have impact on circulation (Pedlosky, 1987). There is a quite substantial literature on observations and theories of large amplitude internal waves in shallow water (Stanton and Ostrovsky, 1998). Many studies concentrate on solitary waves in long wave situations employing the Korteweg-de Vries equation (Holloway et al., 1997), but

not on the highly transient modes with a potential of abrupt growth. For relevance in other fields of physics and engineering, the actual derivation of the governing equations may dictate the regime of input parameter values necessary for rogue waves to occur.

(2) (a) '…*not accessible by a broad, general audience. E.g. What are the initial condition? The authors states that they chose a modulation instability mode with the optimised growth rate. What does this mean? What condition should I use if I want to replicate these results?*…'

Response: The topic of modulation instability has been discussed thoroughly in many widely used monographs in fluid mechanics and optics, e.g.,

Craik, A. D. D.: Wave Interactions and Fluid Flows, Cambridge University Press, 1984.

Mei, C. C.: The Applied Dynamics of Ocean Waves, World Scientific, 1989.

Agrawal, G. P. and Kivshar, Y.: Optical Solitons, Academic Press, 2003.

It may not be fair to the journal to describe all the details of such calculations in this paper. Hence we just add a brief explanation in Section 3 and include the book by Craik:

Modulation instability refers to the growth of small disturbance in a system due to the interplay between dispersive and nonlinear effects (Craik, 1984), and here we examine this by solving the nonlinear Schrödinger equation (Eq. (4)) numerically.

To avoid possible confusion, instead of saying 'proper' or 'optimized' growth rates, we use the words 'maximum' for growth rates and 'baseband modes' in Section 3.

For these types of problems in modulation instability, usually there is a range of unstable wave numbers. The following diagram exhibits the typical growth rate versus the wavenumber:

[Figure]

Our goal is to show that the optimized mode for the generation of rogue wave is a baseband mode at one end of the spectrum with small wavenumber.

(b) '…*The sentence at lines 5-7 at page 4 tries to explain the initial conditions, but it is too technical that general audience would not be able to understand…*'

Response: Numerical simulations with a perturbed plane wave as initial condition have been frequently performed in the literature, starting roughly with say early works on rogue waves about ten years ago (N. Akhmediev, A. Ankiewicz, J. M. Soto-Crespo, *Physical Review E* 80, 026601, 2009). We have also included our own recent work as a reference (Chan, Grimshaw, Chow, *Physical Review Fluids* 2018) in the text, where further details can be found. We choose a special modulation instability mode (one with maximum growth rate) to reduce the time these 'rogue wave like' modes will emerge. The following statement is added in Section 3: This choice of a preferred modulation instability mode as the initial condition is different from other approaches in the literature, such as one using random noise.

(c) '…*Another question was related to the values of the key parameters. Unfortunately, the authors only listed the parameters without providing the requested values, again leaving the reviewer and possible readers unaware of the initial set up of the model…*'

Response: We actually do not understand this request. All the parameter values used in the numerical simulations are given in the figure captions. The corresponding coefficients in the nonlinear Schrödinger equation can readily be computed from Equations (6) and (7). We have added the following sentence in the text (Section 3): Numerical simulations were performed with parameter values appropriate in applications to fluid mechanics.

(3) '…*the manuscript only describes one of condition of density profile with a constant buoyancy frequency $N_0$. …*'

'…*what if N is not constant? This is a relevant question if the goal is to establish the effect of density stratification…*'

Response: The case of constant buoyancy frequency $N$ is simplest and allows analytical formulation. The case 'non-constant' $N$ will be treated in future research effort, as stated near the end of Section 4: Density profiles with variable buoyancy frequency will also be examined in the future.

'…*How do results change if the value of $N_0$ changes? …*'

Response: Simulations with different $N_0$ had actually been given in the revised version of January 2019 (Figure 1). In general, changes in parameter values which enhance a higher 'baseband' modulation instability growth rate will cause the rogue wave to appear sooner. It can be observed from Equations (6) and (7) that the baseband growth rate is independent of $N_0$.

'…*the authors state in the conclusions that constant buoyancy might not play a critical role...*'

Response: We are afraid that those words were taken out of a discussion sequence in an inappropriate manner. The complete description should go like this (Section 4):

'…rogue waves is assessed. Remarkably the constant buoyancy frequency may not play a critical role in the existence condition in terms of focusing, but the mode number of the internal wave does. For breathers or other pulsating modes, this buoyancy frequency parameter will enter the focusing mechanism consideration…'

For rogue waves, $N_0$ will not enter explicitly in the formulation of focusing as long as it is constant. However, it will appear for breathers and other pulsating modes.

(4) '…*I really don't understand the discussion on kh. For a unidirectional sea state, modulational instability ceases for kh < 1.36, although it can survive if oblique modulations are applied (this has been verified down to kh = 1 I guess)…*'

Response: Firstly, the threshold for instability to arise, i.e. $kh > 1.363$, is valid for surface waves only, and we are trying to investigate internal waves now. Secondly, instability for surface wave occurs for sufficiently deep fluid (i.e. $h$ large). For internal waves, we propose that not only is the numerical limit different, but the qualitative picture differs too, i.e. instability occurs for small $h$.

'…*The authors state that internal rogue waves appear for kh < 0.766n\pi. The value of modes n ranges from 1 to 5 (in Table 1). Therefore kh is between 2.4 (for n=1) and 12 (for n=5). These values are considerably larger than the 1.36 threshold and definitely are not of shallow water depth conditions and only marginally consider intermediate water depths. This is very confusing; how can the authors claim that internal rogue waves may arise in shallow and intermediate water regimes, for which kh should be lower than say 2? Maybe I am misunderstanding the manuscript,…*'

Response: We are afraid that the referee had indeed misunderstood the formulation here. The integer refers to the mode number of internal waves. For waves in the interior of the fluid, the vertical structures may have zero, one, two…or integer number of nodes (points of zero displacement), and frequently are referred in the literature as the first ($n = 1$), second ($n = 2$), third ($n = 3$)…mode respectively.

'…*Maybe I am misunderstanding the manuscript, but if this is the case, the authors should make a significant effort to explain their work more comprehensively. If the reviewer cannot understand, it is likely that the broad audience would not understand either…*'

We realize the difficulties in explaining the novel concepts of any paper to a general audience. Anyway we attempt to address a broad audience by adding these remarks about introductory ideas on internal waves in this section:

► Mode number $n$: Internal waves in general display more complex dynamical features than their surface counterparts. As an illustrative example, a given density profile may allow many internal modes characterized by the number of nodes in the vertical structures. This family of allowed states will be generically represented in this paper by an integer $n$ termed mode number.

► Different $kh$ for different $n$: For a wave packet associated with the first internal mode ($n = 1$), modulation instability or rogue wave can occur for carrier wave number $k$ and shallow fluid of depth $h$ in the range of $kh < 0.766\pi$ or 2.406.

► Reinforce this feature for $n$ in Table 1: ([mode number of internal wave, with each]{.underline} [$n$ representing a different vertical structure]{.underline})

(5) '…*Results presented in section 3 are not particularly new to me, at least in a way they are presented*…'

Response: We cannot change the mathematics of the nonlinear Schrödinger equation (NLSE), but the ways and interpretation of NLSE as applied to say fluid mechanics and optics are dramatically different. Here we are trying to illustrate how the same theory of NLSE, as employed for surface and internal waves, yields different interpretations, e.g. in terms of the buoyancy frequency $N$, depth $h$ and internal wave mode number $n$.

'…*striking results that relate to internal wave and deserve a brief communication*…'

Besides the frequently mentioned importance of internal waves in terms of transport as stated earlier, such waves can also play significant roles in underwater acoustics. Hence an abnormally large internally wave appearing abruptly may have physical implications. A remark and references are added in Section 4: Besides their relevance in transport phenomena, internal waves have significant connection with underwater acoustics (Apel et al, 2007; Zhou et al., 1991), and abnormally large internal rogue waves may have physical implications in those aspects.

(6) '…*The authors use an ad-hoc initial condition to achieve maximum instability. Why do the authors claim at line 12 (page 4) that rogue waves emerge spontaneously? It seems to me they are just forcing them to occur and this is far from being spontaneous*…'

Response: We do not agree with the word 'ad-hoc'. The connections between baseband modulation instability and rogue waves from a deterministic approach have been amply demonstrated in the literature (Baronio et al., 2015; Chan et al., 2018 and other works cited there). This choice of preferred mode (baseband mode, Section 3) will allow the rogue wave to emerge sooner than using a random noise as initial condition. One of our goals is to demonstrate the importance of baseband modes in the generation mechanism of internal rogue waves. As the spatial structures of the input and the emerged entities ('rogue wave like' units) differ dramatically, many researchers in this field take this as a demonstration of a 'spontaneous' occurrence.